# Immunodynamic Disruption in Sepsis: Mechanisms and Strategies for Personalized Immunomodulation

**DOI:** 10.3390/biomedicines13092139

**Published:** 2025-09-02

**Authors:** Jhan S. Saavedra-Torres, María Virginia Pinzón-Fernández, Humberto Alejandro Nati-Castillo, Valentina Cadena Correa, Luis Carlos Lopez Molina, Juan Estaban Gaitán, Daniel Tenorio-Castro, Diego A. Lucero Guanga, Marlon Arias-Intriago, Andrea Tello-De-la-Torre, Alice Gaibor-Pazmiño, Juan S. Izquierdo-Condoy

**Affiliations:** 1Facultad de Salud, Universidad Santiago de Cali, Cali 760001, Colombia; 2Grupo de Investigación en Salud (GIS), Departamento de Medicina Interna, Universidad del Cauca, Popayan 190003, Colombia; 3Interinstitutional Group on Internal Medicine (GIMI 1), Department of Internal Medicine, Universidad Libre, Cali 760043, Colombia; 4Facultad de Medicina, Corporación Universitaria Empresarial Alexander von Humboldt, Armenia 630001, Colombia; 5Facultad de Ciencias de la Salud, Universidad del Quindío, Armenia 630001, Colombia; 6One Health Research Group, Universidad de las Américas, Quito 170124, Ecuador; 7Departamento de Ginecología, Hospital Pablo Arturo Suarez, Quito 170702, Ecuador

**Keywords:** sepsis, immunosuppression, T lymphocytes, dendritic cells, apoptosis, biomarkers, sepsis-associated persistent multiorgan immunometabolic syndrome

## Abstract

Sepsis is a life-threatening syndrome caused by a dysregulated host response to infection. It follows a dynamic course in which early hyperinflammation coexists and overlaps with progressive immune suppression, a process best described as immunodynamic disruption. Key mechanisms include extensive lymphocyte death, expansion of regulatory T cells, impaired antigen presentation, and persistent activation of inhibitory checkpoints such as programmed cell death protein 1 (PD-1) and cytotoxic T lymphocyte–associated protein 4 (CTLA-4). These changes reduce immune competence and increase vulnerability to secondary infections. Clinically, reduced expression of Human Leukocyte Antigen–DR (HLA-DR) on monocytes and persistent lymphopenia have emerged as robust biomarkers for patient stratification and timing of immunomodulatory therapies. Beyond the acute phase, many survivors do not achieve full immune recovery but instead develop a Persistent Immune Remnant, defined as long-lasting immune, metabolic, and endothelial dysfunction despite apparent clinical resolution. Recognizing PIR emphasizes the need for long-term monitoring and biomarker-guided interventions to restore immune balance. To integrate these observations, we propose the SIMMP–Sepsis model (Sepsis-Associated Persistent Multiorgan Immunometabolic Syndrome), which links molecular dysfunction to clinical trajectories and provides a framework for developing precision immunotherapies. This perspective reframes sepsis not only as an acute crisis but also as a chronic immunometabolic syndrome, where survival marks the beginning of active immune restoration.

## 1. Introduction

Sepsis is a life-threatening clinical condition that triggers a complex, biphasic immune response. In the early phase, Pathogen-Associated Molecular Patterns (PAMPs) activate pattern recognition receptors (PRRs), such as toll-like receptors (TLRs), initiating a cytokine storm (e.g., TNF-α, IL-1, IL-6). This hyperinflammatory surge drives endothelial dysfunction, vascular leakage, tissue hypoperfusion, and ultimately multiorgan injury [1,2,3,4].

This excessive activation is followed by immune exhaustion and suppression, marked by apoptosis of T cells and antigen-presenting cells, along with reduced expression of HLA-DR on monocytes, which impairs antigen presentation. Concurrently, inhibitory immune checkpoints such as programmed cell death protein 1 (PD-1) and cytotoxic T lymphocyte–associated protein 4 (CTLA-4) are upregulated, while anti-inflammatory cytokines, including IL-10 and TGF-β, dominate the microenvironment, establishing a state of immunoparalysis [5,6,7,8]. These changes increase susceptibility to secondary infections such as ventilator-associated pneumonia, invasive fungal disease, and viral reactivation [1,5,9].

In elderly patients, age-related immunosenescence exacerbates this suppression. Both innate (macrophage and neutrophil activity) and adaptive responses are diminished, prolonging the immunosuppressive phase and contributing to increased late mortality [5,9,10,11,12]. To stratify risk, biomarkers such as reduced monocyte HLA-DR expression and decreased TNF-α production after lipopolysaccharide (LPS) stimulation have been proposed. Immunorestorative interventions—including interferon-γ (IFN-γ), granulocyte-macrophage colony-stimulating factor (GM-CSF), and PD-1 blockade—have shown promise in selected cohorts [1,5,10,12].

A major clinical challenge lies in defining the optimal therapeutic window: early immune stimulation may exacerbate tissue injury, whereas delayed intervention may fail to restore competence. For this reason, sepsis is more accurately described as an evolving immunodynamic process, in which hyperinflammatory and immunosuppressive phases overlap rather than occur sequentially [5,10,12]. Recognizing this dynamic pattern is essential for designing personalized strategies that improve both survival and long-term outcomes.

## 2. Immunodynamic Disruption in Sepsis

Sepsis triggers a complex and dynamic immune response more accurately described as “immunodynamic disruption” rather than simply “immunoparalysis”. This concept captures the fluctuating interplay between uncontrolled inflammation and progressive suppression, avoiding an overly reductionist view of septic immunity.

In the early phase, pathogen recognition via toll-like receptors (TLRs) on innate immune cells induces a cytokine storm—including TNF-α, IL-1β, and IL-6—that drives vascular injury, coagulation activation, and multiorgan dysfunction [1,2,3,4]. Simultaneously, counter-regulatory mechanisms are initiated. As sepsis progresses, the immune system undergoes dynamic reprogramming rather than uniform suppression. This includes regulatory T cell (Treg) expansion, persistent expression of inhibitory checkpoints such as programmed cell death protein 1 (PD-1) and cytotoxic T lymphocyte–associated protein 4 (CTLA-4), and extensive apoptosis of T and B lymphocytes, ultimately reducing adaptive competence [5,10,11,12,13,14,15]. Antigen-presenting cell (APC) dysfunction—evidenced by reduced HLA-DR expression on monocytes and dendritic cell apoptosis—further impairs antigen presentation and increases susceptibility to secondary infections and viral reactivation [5,9,10,11,12]. In addition, low-density neutrophils (LDNs) exhibit impaired phagocytosis and high PD-L1 expression, contributing to tolerance rather than effective clearance [13,14,16,17].

The concept of immunodynamic disruption thus reflects the biphasic, heterogeneous, and evolving nature of immune responses in sepsis. It provides a more precise framework for understanding pathophysiology and guides the design of tailored therapies. Instead of generalized approaches that broadly stimulate or suppress immunity, recognizing this complexity enables personalized interventions according to each patient’s immunological phase and profile [18,19,20,21,22].

To support immunotherapeutic decisions, several biomarkers have been proposed, including surface markers, soluble mediators, and functional indicators of activation or suppression [18,19,20,21,22].

To support immunotherapeutic decisions, several biomarkers have been proposed to assess immune status in septic patients. These include surface markers, soluble mediators, and functional indicators of immune activation or suppression [18,19,20,21,22] (Table 1).

Each alteration has specific implications for monitoring and therapy. Lymphocyte apoptosis underscores the need for regular lymphocyte counts and supports trials with interleukin-7 (IL-7) aimed at restoring adaptive immunity [34,35]. Treg expansion and checkpoint overexpression (PD-1, CTLA-4) justify checkpoint expression as prognostic biomarkers and provide a rationale for selective inhibition [36]. Reduced monocyte HLA-DR expression, validated as a marker of immunoparalysis, guides stratification of patients who may benefit from interferon-γ (IFN-γ) or granulocyte-macrophage colony-stimulating factor (GM-CSF). Neutrophil dysfunction and LDN accumulation support monitoring of PD-L1 and chemotaxis to inform strategies targeting neutrophil survival and function [37]. Finally, evidence of mitochondrial dysfunction and epigenetic reprogramming highlights the value of metabolic biomarkers (e.g., circulating mitochondrial DNA [mtDNA], lactate levels) and supports the potential of metabolic modulators to restore homeostasis [37,38]. Linking mechanisms with measurable biomarkers strengthens translational value and facilitates precision immunotherapy [39].

### Checkpoint Signaling and Treg Expansion

The immunodynamic collapse in sepsis involves profound remodeling of the adaptive system, driven by Treg expansion, checkpoint upregulation, and lymphocyte apoptosis. During the early-to-intermediate phase, Tregs expand disproportionately through persistent IL-2/STAT5 signaling and epigenetic reinforcement of FoxP3 expression [40,41,42]. This skewed repertoire suppresses effector responses and promotes tolerance, a phenomenon also observed in chronic viral infections and tumor microenvironments [43,44,45,46].

Concurrently, T cells upregulate checkpoints such as PD-1 and CTLA-4, which impair T cell receptor (TCR) signaling, reduce cytokine secretion, and hinder pathogen clearance. Experimental models show that sustained checkpoint engagement promotes exhaustion, reduces clonal diversity, and abrogates memory formation [6,47]. These alterations represent critical nodal points in sepsis-induced immunosuppression [23,32].

In parallel, widespread lymphocyte apoptosis—particularly of CD4^+^ and CD8^+^ subsets—depletes the adaptive reservoir, typically mediated via caspase-3 activation and mitochondrial cytochrome c release [26,48]. This triad—Treg-driven suppression, checkpoint-mediated exhaustion, and apoptotic attrition—produces a state of immune collapse marked by impaired antigen-specific responses, nosocomial infections, and viral reactivation. Clinically, this manifests as secondary pneumonia, cytomegalovirus (CMV) or herpes simplex virus (HSV) reactivation, and poor vaccine responsiveness [27,49].

These alterations are not transient. Immunophenotyping studies show persistent T cell anergy and epigenetic scarring weeks after sepsis [14,27,50]. Collectively, these mechanisms provide the rationale for targeted interventions such as IL-7, checkpoint blockade, and caspase inhibitors, which are under evaluation in early-phase trials. This constellation underscores the need for dynamic monitoring and personalized immunointerventions in sepsis survivors [27,50] (Table 1).

## 3. Mechanisms of Immune Dysfunction

The immunological dysfunction observed in sepsis involves both adaptive and innate compartments, affecting multiple cell types through distinct yet overlapping mechanisms. Table 2 summarizes the principal immune cells involved, the key dysregulated pathways, and the resulting clinical consequences.

### 3.1. Adaptive Immune Exhaustion

The immunological dysfunction observed in sepsis involves both adaptive and innate compartments, affecting multiple cell types through distinct yet overlapping mechanisms [25,54]. These checkpoints impair T cell receptor (TCR) signaling, suppress cytokine production (e.g., IL-2, IFN-γ), and limit cytotoxic activity, resulting in a dysfunctional adaptive landscape. Elevated frequencies of PD-1^+^CD4^+^, PD-1^+^CD8^+^, and CTLA-4^+^CD8^+^ cells correlate with higher APACHE II and SOFA scores [25,54].

Mechanistically, PD-1 signaling promotes apoptosis and susceptibility to secondary infections [51,55,56], while Tim-3, through galectin-9 binding, induces tolerance in effector T cells. CTLA-4 dampens activation by competitively blocking CD28 interaction with CD80/CD86 on antigen-presenting cells [25,54]. These suppressive signals converge on mitochondrial apoptotic pathways, including cytochrome c release and caspase-9 activation, leading to profound depletion of the adaptive immune pool [25,54].

Loss of CD4^+^ T cells is consistently associated with poor outcomes, whereas their preservation predicts improved recovery. Notably, blockade of PD-1 or CTLA-4 has demonstrated efficacy in restoring T cell function and improving survival in preclinical sepsis models [25,54].

### 3.2. Innate Effector Failure

Monocytes and macrophages in sepsis exhibit marked downregulation of HLA-DR, impairing antigen presentation and sustaining immune anergy [35,45,46]. Dendritic cells undergo apoptosis and mitochondrial dysfunction [13,14,15,16], with reduced expression of co-stimulatory molecules (CD40, CD80, CD86), diminished IL-12p70 production, and increased IL-10 secretion, which promotes Treg expansion [15,57,58].

Neutrophils initially resist apoptosis and become hyperactivated, releasing proteolytic enzymes and Reactive Oxygen Species (ROS) that exacerbate tissue damage and multiorgan failure [27,59,60]. As sepsis progresses, they develop impaired chemotaxis and phagocytosis due to dysregulated integrin and adhesion molecule expression [31,52,61].

Low-density neutrophils (LDNs), characterized by a CD66b^+^CD63^+^CD11b^+^CD184^+^ phenotype, accumulate within the PBMC fraction. These cells express high PD-L1, resist apoptosis, exhibit poor antimicrobial activity, and suppress T cell proliferation, thereby contributing to increased secondary infections and mortality [15,57,58].

A central driver of innate dysfunction is aberrant activation of the cyclic GMP-AMP synthase (cGAS)–Stimulator of Interferon Genes (STING) pathway, triggered by cytoplasmic mitochondrial DNA (mtDNA) accumulation. This induces type I interferons and IL-10 through JAK–STAT3 signaling [15,57,58]. Experimental inhibition of this axis has been shown to restore dendritic cell function and promote effective T cell responses [30,62,63] (Table 3).

Moreover, early inflammasome activation—marked by ASC-speck^+^ neutrophils and monocytes—is followed by a progressive decline in these cells during late-stage sepsis [30,62,63]. A sustained reduction below 1650 ASC-speck^+^ cells/mL by day six correlates with impaired phagocytosis, defective antigen presentation, and increased 90-day mortality [65,66]. This trajectory, characterized by persistent IL-18 elevation and declining IL-1β levels [65,67,68], reflects a reprogrammed inflammasome phenotype favoring immunosuppression, consolidating immunoparalysis as a hallmark of late-phase sepsis (Figure 1).

## 4. Mechanistic Ambiguities and Translational Targets

Although numerous immunological pathways involved in sepsis have been characterized in both experimental and clinical settings, several remain the subject of conflicting evidence regarding their translational relevance. For example, the PD-1/PD-L1 axis is a well-recognized marker of T cell exhaustion, and its blockade has demonstrated immunorestorative effects in murine models and ex vivo analyses. However, clinical trials in septic patients remain limited and show variable efficacy depending on disease stage and immune profile [33,60,69].

Likewise, NETosis exhibits a dual role: while essential for pathogen containment, excessive Neutrophil Extracellular Trap formation contributes to endothelial injury and thrombosis, complicating its therapeutic targeting [70]. The cGAS–STING pathway presents similar ambiguity—its activation promotes early pathogen sensing, yet persistent signaling, particularly in dendritic cells, fosters immunoparalysis. Thus, the timing and cellular context are critical determinants of its potential as a therapeutic target [29].

In contrast, pathways with more consistent evidence include IL-7–mediated lymphocyte restoration [33,60,69] and β1-adrenergic blockade for immunomodulatory cardiovascular regulation [71]. Both have shown promise in preclinical and early-phase studies and are currently under active investigation as viable therapeutic strategies [33,60,69].

## 5. Persistent Immune Remnant

In sepsis survivors, clinical recovery does not equate to full immunological restoration. Instead, a persistent state of cellular dysfunction emerges, referred to here as the Post-Sepsis Cellular Remnant (PSCR), a pathophysiological entity marked by epigenetic reprogramming, immunometabolic derangements, and maladaptive inflammatory memory. Multiple studies have documented sustained alterations in monocytes (↓HLA-DR), T lymphocytes (↑PD-1, B and T Lymphocyte Attenuator [BTLA]), and expansion of immunosuppressive populations such as myeloid-derived suppressor cells and regulatory T cells (Tregs) [72,73,74,75,76]. These cells demonstrate impaired cytokine production (IL-6, TNF-α, IFN-γ) in response to classical stimuli [18,77,78], alongside mitochondrial dysfunction and disrupted metabolic pathways [79].

### 5.1. Epigenetic Reprogramming

At the epigenetic level, persistent enrichment of histone marks, including H3K9ac, H3K27me3, and H3K4me3, has been observed at regulatory regions of genes such as IL-10, FPR1, HLA-DR, and NLRP3. These modifications persist for weeks after the septic episode and are more pronounced in non-survivors [80,81,82], suggesting a form of non-resolving immune memory directly linked to outcomes [18,77,78].

Recent evidence has also identified histone H3 lactylation at lysine 18 (H3K18la) as a durable epigenetic marker of trained immunity, induced by intracellular lactate accumulation during sepsis [83,84]. This modification promotes open chromatin architecture, enhances proinflammatory gene expression, and persists for up to 90 days post-activation [83,85].

### 5.2. Immunometabolic Alterations

The integration of metabolism and epigenetics is underscored by metabolites such as succinate, fumarate [64,86,87], and acetyl-CoA, which regulate the activity of histone- and DNA-modifying enzymes, establishing stable transcriptional programs in post-septic immune cells [88,89,90].

Accordingly, PSCR manifests as a state of metabolic dysfunction: monocytes exhibit persistent glycolytic activity [91,92], while lymphocytes display fragmented mitochondria and defective biosynthetic capacity [79]. These changes result in dysregulated cytokine release and impaired immune responsiveness [24,93,94].

Functional studies of monocyte-to-macrophage differentiation reveal sustained activation of proinflammatory enhancers enriched in H3K27ac and H3K4me3, marks associated with active transcription and trained immunity [95,96,97]. Unlike tolerance, this programming amplifies responses upon secondary stimulation, reinforcing maladaptive immune memory as a defining feature of PSCR [95,98].

Mitochondrial dysfunction represents a structural cornerstone of PSCR. Cells such as hepatocytes, lymphocytes, and monocytes previously exposed to sepsis exhibit reduced oxygen consumption, membrane depolarization, excessive ROS production, and depleted antioxidant reserves [28,38,99]. These abnormalities can be partially reversed by activating the AMPK–PGC-1α–autophagy axis, as shown in preclinical models with ginsenoside Rg3 [100,101,102,103]. Low expression of PGC-1α, a master regulator of mitochondrial biogenesis, perpetuates oxidative stress and sustains a proinflammatory basal state [100,101,103].

### 5.3. Endothelial Dysfunction and Immunothrombosis

At the endothelial level, sepsis leaves a pathological imprint characterized by persistent activation, with elevated angiopoietin-2, soluble vascular cell adhesion molecule-1 (sVCAM-1), endothelial microparticles, and impaired vascular integrity [104,105,106]. This dysfunction promotes a microenvironment conducive to immunothrombosis, involving neutrophil extracellular trap (NET) formation [105,107,108], tissue factor activation, and persistence of a procoagulant state even after clinical resolution [52,109].

Persistent endothelial activation is thus an integral component of PSCR, contributing to progressive vascular deterioration and chronic tissue hypoperfusion [105,107,108]. Collectively, these structural, functional, and metabolic alterations define PSCR as a clinically relevant post-inflammatory immunometabolic state [72,73,74,75,76]. This state is measurable through specific cellular and molecular biomarkers and carries significant implications for post-ICU monitoring, late infection risk, and long-term mortality [72,110]. Recognizing PSCR demands a paradigm shift in sepsis follow-up—from a survival-centric model to one focused on long-term immunometabolic recovery and reprogramming of innate immunity as therapeutic goals [111,112] (Table 4).

### 5.4. Clinically Relevant Biomarkers of PSCR

Several biomarkers have emerged as predictors of immune dysfunction and adverse outcomes in sepsis survivors. Reduced HLA-DR expression on monocytes remains a key indicator of immunosuppression and correlates with increased risk of secondary infections and mortality [89,90,91].

Persistently elevated IL-6 during recovery is associated with ongoing lymphocyte dysfunction and poor long-term prognosis. Additional markers such as IL-10 and soluble TNF receptor 1 (sTNFR1) hold independent prognostic value, particularly in patients with protracted courses [92,93,94].

At the transcriptomic level, sustained upregulation of inhibitory receptors, including PD-1 and BTLA, on CD4^+^ and CD8^+^ T cells reflects an exhausted phenotype, predisposing to opportunistic infections and viral reactivations [95,96].

Collectively, these biomarkers provide mechanistic insights into persistent post-sepsis immunopathology and represent clinically actionable tools for patient stratification and the development of personalized immunomodulatory strategies during recovery [18,86,87].

## 6. Experimental Models of Immunodysfunction

### 6.1. Classical Animal and In Vitro Models

Experimental models have been central to advancing the understanding of immunoparalysis in sepsis. One of the most commonly used approaches is the one-hit model, which induces sepsis through a single insult—such as administration of lipopolysaccharide (LPS), injection of live bacteria, or surgical infection (most commonly via cecal ligation and puncture [CLP]) [65,66]. This model enables the study of early immune alterations, including lymphocyte apoptosis and the expansion of immunosuppressive populations like Tregs and MDSCs. However, it fails to replicate the temporal progression from hyperinflammation to sustained immunosuppression typically seen in human sepsis [65,66,113].

To address this limitation, the two-hit model was developed. Here, an initial septic event is followed by a secondary infection, commonly with pathogens such as *Pseudomonas aeruginosa* or *Candida albicans* [65]. This approach more accurately reflects the clinical course of sepsis in humans, reproducing features such as impaired resolution of inflammation, reduced lymphocyte proliferation, and elevated expression of inhibitory receptors. Despite its improved translational relevance, standardization remains challenging due to variability in the type and timing of the secondary insult [65,67,68].

Another widely studied method is the LPS tolerance model, which uses repeated sublethal doses of LPS to induce a hyporesponsive state in leukocytes [67,68]. This model simulates key features of immune reprogramming, including diminished cytokine production and functional suppression of innate responses. However, because it relies solely on endotoxin exposure and not on polymicrobial infection, it does not fully reflect the complexity of human sepsis [65,68].

To increase clinical relevance, newer models have incorporated aged animals, mice with comorbidities (e.g., obesity, cancer), or “dirty mice” exposed to natural microbiota. These modifications better capture the immunological heterogeneity observed in patients and demonstrate how baseline characteristics shape the trajectory of sepsis-induced immunosuppression. At the same time, they introduce additional variability that complicates reproducibility and cross-study comparison [65,114,115].

Despite these limitations, murine models remain essential tools in the study of immunoparalysis. They have facilitated the identification of key biomarkers and informed the development of targeted therapies. Nonetheless, the gap between preclinical models and human immune complexity underscores the urgent need for improved reproducibility, standardization, and translational alignment [113,114] (Table 5).

### 6.2. Advanced and Translational Platforms

The elucidation of mechanisms underlying the Post-Sepsis Cellular Remnant (PSCR) has been supported by a diverse array of experimental models, each offering distinct methodological advantages and varying degrees of translational relevance.

The murine CLP model has been widely used to characterize sepsis-induced lymphocyte apoptosis and mitochondrial dysfunction. However, its translational value remains limited, as it does not replicate the complex epigenetic reprogramming observed in human sepsis survivors [116,117,118].

In contrast, ex vivo models using human monocytes have proven highly informative for uncovering persistent metabolic rewiring and stable epigenetic marks characteristic of PSCR. These models offer strong clinical relevance but are limited by the inability to support longitudinal, patient-specific follow-up.

Transcriptomic analyses—particularly RNA-seq and single-cell RNA-seq in critically ill patients—currently represent the translational gold standard. These technologies provide high-resolution characterization of immune phenotypes during and after sepsis [93,119,120]. Nevertheless, their application is restricted by high cost and the critical need for precisely timed sampling.

Meanwhile, 3D endothelial organoid models have shown value in studying chronic inflammation and vascular barrier dysfunction. However, their translational applicability remains moderate due to the absence of integrated immune cell interactions. Taken together, these platforms should be regarded as complementary rather than exclusive, each contributing to a more comprehensive understanding of PSCR pathophysiology [121,122,123].

### 6.3. Strategies to Enhance Translational Relevance of Experimental Models

Although the limitations of experimental models have been extensively discussed, bridging the gap with human sepsis requires clear strategies to enhance translational value. Current recommendations emphasize the adoption of standardized protocols, such as the MQTiPSS guidelines, to reduce variability across laboratories. In addition, the incorporation of clinically relevant variables—including age, sex, and comorbidities—into animal models is essential to reflect patient heterogeneity. The use of more complex systems, such as humanized or “dirty” mice exposed to natural microbiota, further improves biological relevance by better approximating human immune diversity [124]. Equally important is the alignment of experimental endpoints with clinically meaningful outcomes, including immune cell exhaustion, biomarker dynamics, and long-term mortality. Finally, harmonization of reporting standards and the establishment of shared data repositories are necessary to improve reproducibility and facilitate direct benchmarking across studies [125,126]. Together, these measures strengthen the translational relevance of preclinical research and accelerate the development of clinically applicable immunotherapies [124].

Bridging the gap between preclinical and clinical research requires aligning model-derived findings with patient data. For instance, lymphocyte apoptosis and HLA-DR downregulation in cecal ligation and puncture models mirror changes consistently seen in septic patients. Likewise, LPS tolerance models reproduce impaired TNF-α production and monocyte dysfunction validated in human cohorts. Such parallels reinforce the translational value of experimental models and support biomarker-based approaches that can be prospectively tested in clinical settings.

## 7. Adaptive Immune Collapse

### 7.1. Septic Lymphocytic Apoptosis

Lymphocyte apoptosis in sepsis represents one of the most profound and well-characterized disruptions of the adaptive immune system. Clinical and post-mortem studies consistently demonstrate a massive depletion of B lymphocytes and CD4^+^ T cells during the acute phase, primarily via mitochondrial apoptotic pathways involving caspase-9 [1,2]. Paradoxically, this occurs during active infection, when clonal expansion would normally be expected to reinforce host defense.

Human spleen analyses reveal that not all subsets are equally affected. CD8^+^ T cells and natural killer (NK) cells are relatively preserved, whereas B cells and CD4^+^ T cells—particularly follicular helper T cells (Tfh)—are severely reduced. Loss of Tfh cells compromises humoral immunity by disrupting germinal center formation, memory B cell development, and long-term antibody production [1,2].

Surviving lymphocytes often exhibit functional exhaustion, characterized by diminished cytokine secretion, increased expression of inhibitory receptors such as programmed PD-1, and impaired proliferative capacity. These alterations contribute to a state of adaptive immunosuppression that may persist for weeks after clinical resolution. From an immunodynamic perspective, apoptosis is not a passive consequence but an active driver of adaptive immune collapse. Early identification through immunophenotyping or tissue sampling may support risk stratification and guide targeted interventions such as IL-7 therapy, PD-1 blockade, or restorative strategies for B and CD4^+^ T cells [1,2].

### 7.2. Septic Treg Suppression

Regulatory Treg expansion in sepsis is marked by a highly suppressive phenotype defined by CD25^high and FoxP3^+^ expression, together with upregulation of inhibitory molecules including cytotoxic T lymphocyte–associated protein 4 (CTLA-4) and PD-1 [3,4,5]. These cells secrete Interleukin-10 (IL-10) and transforming growth factor beta (TGF-β), sustaining a profoundly immunosuppressive milieu.

In this context, dendritic cell maturation is impaired, CD4^+^ and CD8^+^ T cell proliferation is suppressed, and the phagocytic activity of macrophages and NK cells is diminished. Moreover, imbalance of the Th17/Treg axis—evidenced by reduced IL-17A and IL-22 production—further weakens mucosal and systemic defenses against bacterial and fungal infections [3,4]. Clinically, an elevated Treg/Th17 ratio correlates with higher Sequential Organ Failure Assessment (SOFA) scores, persistent infection beyond day 7, and increased 28-day mortality [3,4,5].

Tregs therefore act as both markers and drivers of immune dysregulation. Serial monitoring (CD4^+^CD25^highFoxP3^+^/CD127^low) and selective modulation via PD-1 inhibitors or IL-10 receptor blockade are emerging as promising strategies to recalibrate immune balance without triggering autoimmunity [3,4,5].

### 7.3. Adaptive Immune Resuscitation

Recent clinical trials with recombinant human IL-7 (CYT107) and the PD-1 inhibitor nivolumab have advanced personalized immunotherapy in sepsis. The IRIS-7 study demonstrated that IL-7 is safe and effective in patients with septic shock and severe lymphopenia, inducing a three- to four-fold increase in CD4^+^ and CD8^+^ T cell counts, enhancing proliferation, and restoring functional activity without precipitating cytokine storms or organ failure [88,89,90].

Similarly, nivolumab was well tolerated in patients with overt immunosuppression (lymphocyte count < 1.1 × 10^3^/μL). Although the phase 1b trial did not assess efficacy, findings showed increased monocyte HLA-DR expression, sustained PD-1 receptor occupancy (>90% for at least 28 days), and no increase in pro-inflammatory cytokines. These results support checkpoint blockade as a viable strategy to restore T cell function and counteract immune paralysis [88,89,90].

Taken together, IL-7 and PD-1 blockade represent complementary approaches that may reverse immunodynamic disruption, reduce secondary infections, and prevent progression toward persistent immune dysfunction [88,89,90].

### 7.4. Sepsis Immune Phases

Sepsis is a dynamic condition in which pro-inflammatory and immunosuppressive pathways coexist, with their relative dominance determining clinical trajectory. In the early phase, hyperinflammation prevails, driven by massive cytokine release, endothelial activation, and multiorgan dysfunction [60]. Clinically, this phase manifests with fever, hemodynamic instability, and elevated acute-phase reactants.

In survivors of the initial insult, the immune response often shifts toward sustained immunosuppression. This phase is characterized by persistent lymphopenia, impaired TNF production after ex vivo stimulation, reduced monocyte HLA-DR expression, and increased susceptibility to secondary infections or viral reactivation [60,127]. Mechanistically, defects in both innate and adaptive immunity underlie this state, including apoptosis of effector cells, Treg expansion, T cell exhaustion, accumulation of myeloid-derived suppressor cells, and loss of co-stimulatory signaling [60,127].

Distinguishing between these phases requires integrating clinical features with laboratory parameters. Recognition of this temporal and mechanistic complexity is essential for tailoring interventions that avoid exacerbating inflammation in the early stage while restoring immune competence in the later phase [60,127].

## 8. SIMMP–Sepsis: Toward an Integrative Framework

Current understanding of sepsis increasingly recognizes that it extends beyond the classical definition of acute organ dysfunction triggered by infection [128,129]. Growing evidence shows that, in many patients, sepsis initiates a persistent immunometabolic syndrome that lasts well beyond apparent clinical recovery [96,98,130,131,132,133]. Nevertheless, most of the literature still fragments this process into acute and post-acute phases, lacking a unifying framework to explain their continuity.

To address this gap, we propose the SIMMP–Sepsis model (Sepsis-Associated Persistent Multiorgan Immunometabolic Syndrome), a conceptual framework that delineates five interconnected levels linking the initial immune response to the chronic dysfunction observed in survivors [96,98,131].

The first level represents the acute activation phase, initiated by recognition of PAMPs and DAMPs and characterized by hyperinflammation, NETosis, complement activation, mitochondrial injury, oxidative stress, and endothelial dysfunction. These processes converge to precipitate multiorgan failure that typically requires intensive support. However, resolution of this phase does not equate to true immunological recovery [132,134,135].

The second level corresponds to persistent cellular reprogramming during the post-critical period. Immunologically, it encompasses trained immunity and maladaptive tolerance, orchestrated by epigenetic modifications such as histone lactylation and H3K4me3 enrichment [123,135]. Metabolically, dysregulated pathways resembling the Warburg effect—marked by lactate and succinate accumulation and impaired tricarboxylic acid (TCA) cycle activity—sustain a pro-inflammatory environment and hinder the restoration of homeostasis [136,137,138].

The third level reflects sustained organ dysfunction. The endothelium remains activated, with increased expression of ICAM-1 and VCAM-1 and compromised vascular integrity. Simultaneously, mitochondrial fragmentation, incomplete mitophagy, and mitochondrial DNA (mtDNA) release exacerbate dysfunction [98,139,140,141]. In the central nervous system, persistent microglial activation maintains neuroinflammation [134]. At the systemic level, the gut–liver–brain axis is disrupted by dysbiosis, microbial translocation, and reduced short-chain fatty acid (SCFA) production, contributing to ongoing hepatic and systemic inflammation [98,139].

The fourth level represents the clinical manifestations of this biology, which include fatigue, myalgias, dysautonomia, cognitive decline, recurrent infections, and fibrotic remodeling across multiple organs [128,142]. These post-sepsis symptoms are often underdiagnosed and misinterpreted as isolated sequelae, rather than as evidence of persistent immunometabolic dysregulation. This highlights the need for long-term monitoring strategies capable of identifying and addressing sustained immune dysfunction [127,133,142].

Finally, the fifth level consolidates SIMMP–Sepsis as a distinct clinical entity, defined by the integration of immune, metabolic, vascular, neurological, and microbial alterations into a persistent syndrome. This framework provides a foundation for risk stratification, biomarker development, and the design of tailored immunometabolic interventions that extend beyond Intensive Care Unit (ICU) discharge [96,98,130,131,132,133].

Rather than replacing classical definitions of sepsis [143], SIMMP–Sepsis expands the conceptual framework by linking molecular mechanisms with clinical outcomes and therapeutic opportunities. This perspective promotes a shift toward precision immunology aimed not only at ensuring survival but also at restoring long-term immune and metabolic integrity [96,98,130,131,132,133] (Table 6).

## 9. Clinical Implications of Sepsis-Induced Immunosuppression

### 9.1. Sepsis and Cancer Immunosurveillance

Sepsis induces a prolonged state of immunoparalysis that compromises both antimicrobial defense and tumor immunosurveillance. A key consequence is dysfunction of CD8^+^ T cells, which are central to identifying and eliminating malignant cells through cytokine secretion and cytotoxic activity [130,144]. Normally, these lymphocytes infiltrate tumor tissue, secrete IFN-γ, and engage in direct cytolytic killing. However, following sepsis, their number and functionality are markedly reduced. Expression of receptors such as PD-1 and LAG-3, necessary for effector function and retention within the tumor microenvironment, is also diminished [144,145,146].

In addition, sepsis impairs CD8^+^ T cell renewal and persistence in tissues. This coincides with reduced MHC-I expression on tumor cells, limiting antigen recognition and facilitating immune escape. Consequently, tumor progression accelerates in sepsis survivors, who face higher risks of de novo malignancies and worse trajectories of preexisting cancers [144,145,146]. Checkpoint inhibitor therapy may restore CD8^+^ T cell function in this context, but its efficacy depends on timely administration. Early intervention—before extensive lymphocyte depletion—can reestablish cytotoxic function and limit tumor growth. In contrast, delayed treatment is less effective due to irreversible immune depletion, underscoring the need for early immune profiling and targeted restoration strategies [50,144,146].

### 9.2. Pulmonary Immune Reprogramming

Long-term sepsis survivors frequently exhibit altered pulmonary immunity, which predisposes them to exaggerated inflammatory responses and increased susceptibility to lung injury. Upon subsequent exposure to LPS, these individuals display amplified cytokine release, enhanced neutrophilic infiltration, and aggravated alveolar damage mediated by overactive Ly6C^hi monocytes producing elevated TNF-α [147,148]. Notably, this primed state persists for weeks after clinical recovery and occurs even without active infection, reflecting a failure to restore pulmonary immune homeostasis.

This immune priming involves persistent epigenetic and metabolic reprogramming in myeloid cells. Murine models show that three weeks post-cecal ligation and puncture (CLP), lungs retain expanded Ly6C^hi monocyte populations and sustain inflammatory gene expression even in the absence of additional stimuli. Secondary LPS exposure triggers severe lung injury, including increased permeability, epithelial apoptosis, and excessive NLRP3 inflammasome activation with elevated TNF-α, IL-1β, and IL-6 [147,148,149].

At the molecular level, prolonged leukocyte expansion and persistently inflammatory transcriptomic profiles reinforce a “primed” immune state that coexists with systemic immunoparalysis, predisposing survivors to sterile inflammatory damage [147,150,151]. DAMPs such as S100A8/A9 (Calprotectin) remain elevated, promoting chronic inflammation and increasing vulnerability to secondary insults. In clinical studies, elevated plasma S100A8/A9 correlates with pulmonary complications, highlighting its translational relevance [147,148,152].

Circulating biomarkers such as S100A8/A9 therefore represent both early indicators and potential therapeutic targets for personalized interventions. Post-sepsis surveillance should incorporate dynamic immune profiling with emphasis on pulmonary-specific pathways. This challenges the conventional notion of uniform post-sepsis immunosuppression and supports a more nuanced, stratified, and proactive model of long-term care [150,151].

### 9.3. Calprotectin Endothelial Disruption

Calprotectin (S100A8/A9) is a heterodimeric protein complex released in large amounts by neutrophils and monocytes during intense inflammatory responses such as sepsis. Beyond serving as a biomarker, Calprotectin exerts direct pathogenic effects on the endothelium by inducing mitochondrial disruption. In endothelial cells, it interferes with components of mitochondrial respiratory chain complex I, reducing oxidative activity and the NAD^+^/NADH ratio. This suppresses the activity of Sirt1, a deacetylase that regulates mitochondrial biogenesis, fusion, fission, and degradation [153].

As a result, mitochondria undergo excessive fragmentation, mitophagy is blocked, and damaged organelles accumulate in the cytoplasm. This mitochondrial overload culminates in the release of mtDNA, which activates cytosolic receptors such as ZBP1, initiating PANoptosis—a form of programmed cell death integrating apoptosis, pyroptosis, and necroptosis. These processes destroy functional endothelium, exacerbate microvascular injury, and compromise tissue perfusion [153,154].

### 9.4. IL-36, Sepsis, and Lung Injury

Sepsis, a syndrome defined by a dysregulated host response to infection, disrupts immune homeostasis and contributes to multi-organ failure, notably in the lungs [155,156]. In this context, IL-36 cytokines—IL-36α, IL-36β, and IL-36γ—have been identified as key modulators of inflammation and epithelial integrity [155]. Their serum concentrations increase during the early septic phase, suggesting a role in the initial immune response; however, their precise contribution to the pathophysiology of sepsis and lung injury remains incompletely elucidated [74,75,76].

Clinically, elevated IL-36 levels correlate with sepsis severity and may serve as prognostic biomarkers. Lower IL-36 concentrations in non-survivors imply a protective immunomodulatory role, potentially through regulation of inflammatory cascades and preservation of pulmonary architecture [155,156].

The IL-36 receptor (IL-36R), expressed predominantly on epithelial cells and pulmonary fibroblasts, is essential for orchestrating local immune responses during systemic inflammation [155,156]. In CLP-induced sepsis models, IL-36R deficiency leads to impaired pathogen clearance, increased bacterial load, and aggravated organ damage. This is associated with diminished production of lipocalin-2 (LCN2), an antimicrobial protein that restricts bacterial proliferation in the lungs, thereby facilitating microbial dissemination [155,156,157].

Moreover, IL-36R deficiency exacerbates alveolar epithelial apoptosis, compromising barrier integrity and promoting neutrophil infiltration. This cascade culminates in sepsis-associated lung injury characterized by increased vascular permeability, alveolar edema, and impaired gas exchange [155,157].

### 9.5. Trained Immunity and IL-4

Trained immunity—the epigenetic and metabolic reprogramming of innate immune cells such as monocytes—offers a promising strategy to reverse sepsis-induced immunoparalysis [156,157]. IL-4 plays a dual immunomodulatory role [158,159]. Initially, it dampens inflammation by suppressing the production of proinflammatory cytokines such as TNF-α and IL-6 in activated monocytes [160,161], primarily via STAT6-dependent pathways that promote an anti-inflammatory macrophage phenotype [156,157].

Paradoxically, IL-4 also induces a long-lasting form of trained immunity through PI3K–mTOR signaling, enhancing monocyte responsiveness to secondary challenges. Upon restimulation, these reprogrammed cells produce elevated levels of IL-6 and TNF-α, thereby restoring innate immune competence and counteracting endotoxin-induced hyporesponsiveness [156,157].

To minimize systemic effects and improve delivery, an innovative nanotherapy has been developed [158,159]. This platform fuses IL-4 to apolipoprotein A1 (apoA1) and incorporates it into lipid nanoparticles, enabling targeted delivery to hematopoietic tissues such as the spleen and bone marrow. In preclinical models, this approach restored proinflammatory cytokine production and enhanced adaptive responses, effectively reversing immunoparalysis [156,157].

### 9.6. β1-Adrenergic Modulation

Sympathetic overactivation during sepsis promotes β1-adrenergic signaling, which amplifies Treg-mediated immunosuppression and attenuates CD4^+^ T cell proliferation. This immunologic shift is associated with reduced host capacity to control secondary infections and an increased risk of nosocomial complications [51,55]. β1-adrenergic stimulation exacerbates Treg-driven inhibition, suppressing proinflammatory responses and limiting effector T cell activation. Consequently, adrenergic signaling influences both cardiovascular dynamics and immune homeostasis in sepsis [51,55,56].

Pharmacological blockade of β1 receptors with esmolol [162], or genetic ablation in murine sepsis models, has been shown to reverse these immunosuppressive effects. Such interventions restore CD4^+^ T cell proliferation, reduce Treg frequency, and rebalance the immune response, as evidenced by decreased levels of both pro- and anti-inflammatory mediators [51,55,56]. Notably, β1 inhibition preserves cardiac efficiency despite heart rate reduction, thereby improving tissue perfusion without compromising hemodynamic stability [51,55,56].

### 9.7. Tachycardia and Immune Regulation

Tachycardia in sepsis reflects persistent sympathetic drive in response to systemic inflammation and hypoperfusion. While initially compensatory, sustained catecholaminergic stimulation impairs myocardial relaxation, reduces diastolic filling and coronary perfusion, and exacerbates septic cardiomyopathy through oxidative stress and contractile dysfunction [51,55,56].

This cardiovascular compromise further impairs immune responses by disrupting metabolic balance and reducing perfusion to immunologically active organs such as the spleen and bone marrow [51,163]. Simultaneously, adrenergic signaling promotes immune dysfunction by inducing inhibitory receptors—including PD-1, CTLA-4, and Tim-3—on T cells and monocytes [51,55,56]. This contributes to lymphocytopenia, reduced cytokine production, antigen-presenting cell dysfunction, and pathogen persistence—hallmarks of late-phase sepsis immunoparalysis [14,51,71,164,165].

Therapeutic modulation of heart rate offers dual benefits. β-blockers such as esmolol and landiolol stabilize hemodynamics and partially restore immune function by enhancing lymphocyte activity [162]. Ivabradine, a selective sinoatrial node inhibitor, lowers heart rate without affecting contractility, improving perfusion of immune-relevant tissues [14,51,71].

Together, these interventions emphasize the pathophysiological link between cardiovascular and immune dysfunction in sepsis. Heart rate modulation serves not only to optimize cardiac performance but also to promote immune recovery and reduce sepsis-related morbidity and mortality [14,51,71,164,165].

## 10. Ebola-Driven Immunoparalysis

The study of immune dysregulation in viral sepsis—particularly in infections caused by highly virulent pathogens such as the Ebola virus—reveals a dual profile of inflammatory hyperactivation combined with profound immunosuppression. Clinically, this presents as severe lymphocytopenia, loss of immune surveillance, progressive endothelial dysfunction, and multiorgan failure. At the cellular level, infected cells release large amounts of pyrogenic cytokines (IL-1β, TNF-α, IL-18) and DAMPs, which induce bystander cell death in uninfected immune cells through indirect inflammatory mechanisms. This process leads to apoptosis, functional exhaustion, or loss of viability in monocytes, T lymphocytes, and endothelial cells, independently of direct viral replication [166,167,168].

Within this inflammatory milieu, effector cells exhibit impaired cytotoxicity, reduced IFN-γ production, and diminished antigen-specific responses. In parallel, persistent metabolic and epigenetic alterations sustain an abnormal immune profile beyond the acute phase, aligning with the concept of a dysfunctional post-sepsis remnant characterized by chronic inflammation and progressive immune impairment [166,167,168]. Exaggerated inflammatory signaling also hampers lymphocyte regeneration and promotes the coexistence of both hyperinflammatory and immunosuppressive phenotypes, perpetuating immune paralysis. These mechanisms have been validated in vitro and are consistent with histopathological observations in patients with severe viral sepsis, including lymphoid tissue disruption, thymic atrophy, and widespread endothelial injury [166,167,168].

These insights provide a rationale for targeted immunotherapeutic approaches. Experimental inhibition of inflammasome activation, blockade of IL-1β, and modulation of non-apoptotic cell death pathways have shown beneficial effects, including cellular preservation, restoration of immune function, and reduction of systemic inflammation. Such strategies highlight the potential of phase-specific immunomodulation in viral sepsis [166,167,168].

The persistence of lymphocytopenia, residual inflammation, and endothelial dysfunction after resolution of acute viral sepsis underscores that sepsis is not a transient inflammatory event but a syndrome of sustained immunometabolic dysfunction. This post-inflammatory state increases susceptibility to reinfections and long-term complications, emphasizing the need for early recognition, longitudinal monitoring, and proactive immunologic management [166,167,168].

Overall, these findings strengthen the clinical and mechanistic framework of immunodynamic disruption in sepsis. They demonstrate that unresolved inflammation, secondary immune cell loss, and persistent epigenetic imprinting act as pathogenic cycles in Ebola-driven immunoparalysis, offering functional evidence to guide the design of personalized immunotherapeutic strategies aimed at restoring immune competence and preventing chronic inflammation induced by highly virulent viral pathogens [166,167,168].

## 11. Post-Inflammatory Oncologic Surveillance

A binational cohort study conducted in Finland and Sweden followed over 18,500 sepsis survivors for at least one year, revealing a significantly increased incidence of cancer compared to the general population. With a median follow-up of 3.5 years, standardized cancer incidence rates were markedly elevated in both men and women, particularly for gastrointestinal, urinary tract, and skin malignancies. These findings support the hypothesis that persistent immune dysfunction after sepsis contributes to oncogenesis through prolonged loss of immune surveillance and unresolved chronic inflammation [169,170].

Complementary evidence from a large epidemiological study based on U.S. healthcare records of older adults demonstrated a significantly elevated risk of developing at least fifteen distinct types of cancer following sepsis. This association persisted after adjusting for age, sex, comorbidities, and hospitalization frequency. The most frequently observed malignancies included lung, colorectal, bladder, and pancreatic cancers. Proposed mechanisms include sustained immunosuppression, epigenetic reprogramming, and microbiome dysbiosis, which may collectively facilitate malignant transformation [169,170].

These findings underscore a growing recognition of the link between sepsis and subsequent cancer development, reinforcing the need for proactive oncologic surveillance in post-sepsis care. Surviving sepsis may entail a significantly increased medium-term cancer risk, warranting implementation of early screening programs, risk stratification protocols, and preventive interventions as integral components of post-sepsis management [169,170].

## 12. Precision Immunotherapy After Sepsis

The use of immune checkpoint inhibitors in the post-sepsis setting has attracted growing interest; however, their efficacy depends strongly on timing and careful patient selection [171,172]. Preclinical studies in animal models demonstrate that delayed administration of anti–PD-1 antibodies during the immunosuppressive phase significantly improves survival, restores T cell function, and enhances pathogen clearance. These findings underscore that administration during the early hyperinflammatory phase may be ineffective or even harmful, whereas targeted use in the context of sustained immune dysfunction can yield meaningful clinical benefits [173,174].

Early-phase clinical trials with checkpoint inhibitors, such as nivolumab and anti–PD-L1 antibodies, have confirmed safety in patients with severe sepsis and established immunosuppression. Nonetheless, efficacy results remain inconsistent, highlighting the need for refined patient stratification. Rather than focusing exclusively on therapeutic blockade, these studies emphasize the importance of identifying reliable biomarkers—such as persistent lymphopenia, sustained PD-1 expression on T cells, and reduced HLA-DR levels on monocytes—to define the optimal therapeutic window. Incorporating these parameters into trial design is essential to move beyond safety assessment and determine whether precision immunotherapy can improve clinical outcomes [173,174].

Thus, post-sepsis immunotherapy should be conceptualized as a personalized strategy, guided by dynamic biomarkers and adapted to specific therapeutic windows [171,172]. Only a stratified approach can effectively restore immune competence while minimizing the risk of exacerbating inflammation or disrupting systemic homeostasis [173,174] (Table 7).

## 13. Discussion

Sepsis is increasingly recognized not as a transient inflammatory event but as a persistent immunometabolic disorder with overlapping phases of hyperactivation and suppression [164]. While classical models described a linear progression from cytokine storm to immunoparalysis, our synthesis supports a more dynamic view in which immune activation, exhaustion, and tolerance coexist across anatomical and temporal compartments. This reconceptualization aligns with the concept of immunodynamic disruption, which provides a more integrative framework than the biphasic paradigm and accounts for long-term sequelae driven by epigenetic and metabolic reprogramming.

To bridge the gap between acute immune collapse and chronic dysfunction, we proposed the SIMMP–Sepsis framework. This model delineates five interconnected levels—ranging from acute activation to persistent clinical phenotypes—thereby integrating molecular events such as checkpoint overexpression (PD-1, CTLA-4), histone lactylation, and mitochondrial dysfunction with post-sepsis manifestations including recurrent infections, cognitive decline, and organ fibrosis [92,115,124]. As illustrated in Figure 2, SIMMP–Sepsis highlights the added value of dynamic immune profiling for precision interventions.

This conceptual evolution is summarized in Table 8, which contrasts classical and emerging models of septic immunopathology and highlights the added value of immunodynamic disruption in guiding biomarker-informed precision interventions.

Our analysis also contextualizes well-established mechanisms—including CD4^+^ and B cell apoptosis, Treg expansion, dendritic cell dysfunction, and neutrophil tolerance—within this longitudinal trajectory. Importantly, we identify the persistence of biomarkers such as S100A8/A9, IL-10, mtDNA, and histone lactylation as molecular signatures of a Post-Sepsis Cellular Remnant. Experimental evidence from two-hit and LPS tolerance models reinforces these findings, though translational gaps remain due to differences between murine and human immune complexity [65].

Therapeutic implications emerge from this framework. Promising interventions such as IL-7, PD-1 blockade, GM-CSF, β1-adrenergic inhibition, and metabolic modulators must be evaluated not as universal treatments but as phase-specific strategies guided by longitudinal biomarker monitoring (e.g., HLA-DR, PD-1, IL-6, mtDNA, S100A8/A9). In this regard, precision immunotherapy depends less on the availability of agents than on accurate patient stratification and timing [25,47,82,152]. The inconclusive evidence surrounding IL-36 exemplifies the need for cautious interpretation of exploratory mediators until validated in larger, controlled cohorts [155,175].

We believe the SIMMP–Sepsis model provides a testable and adaptable framework for advancing immunotherapeutic strategies into clinical practice—not as a replacement for current definitions but as a translational bridge linking molecular dysfunction to actionable interventions. Its validation requires prospective and longitudinal application in real-world cohorts, with integration of epigenetic, metabolic, and immunophenotypic markers to stratify patients according to their immune trajectory and therapeutic needs. In parallel, machine learning approaches hold promise for identifying sepsis endotypes with differential responses to treatment, thereby facilitating precision immunotherapy. Moreover, targeting metabolic pathways—such as enhancing mitochondrial biogenesis via PGC-1α activation or reducing lactate-driven immunosuppression—represents an innovative avenue to restore immune competence in sepsis survivors.

Ultimately, the central barrier is not the scarcity of mechanistic insights but the absence of conceptual integration. By framing sepsis as a persistent immunometabolic syndrome, the SIMMP–Sepsis model provides a multidimensional scaffold that links basic immunology with clinical practice. This perspective reframes post-ICU care, emphasizing that survival marks not the end of treatment but the beginning of long-term immune restoration.

### Limitations

This work was conceived as a comprehensive review integrating mechanistic, experimental, and clinical perspectives on immunodynamic disruption in sepsis. Unlike systematic reviews, comprehensive reviews do not follow standardized frameworks such as PRISMA but instead aim for a broad synthesis of available evidence. While this allows the inclusion of diverse data—from preclinical models to emerging conceptual frameworks, it also entails inherent limitations.

First, the absence of a rigid methodology increases the risk of selection bias. Although targeted searches in PubMed, Scopus, and Web of Science were performed using specific keywords, the narrative approach cannot guarantee exhaustive coverage, and some relevant studies, particularly preprints or non-English publications, may have been missed.

Second, heterogeneity across studies limits comparability. Differences in study design, patient populations, timing of sampling, and laboratory techniques (e.g., HLA-DR measurement, transcriptomic protocols) hinder direct integration. Similarly, preclinical models such as cecal ligation and puncture or LPS tolerance capture only partial aspects of sepsis and do not fully reflect human immunology, restricting translational value.

Finally, this review prioritizes breadth over quantitative rigor. No formal risk-of-bias assessment or meta-analysis was conducted; instead, emphasis was placed on convergent findings, mechanistic plausibility, and translational implications.

Despite these limitations, the review provides an updated, integrative perspective linking molecular mechanisms, clinical implications, and therapeutic strategies, while proposing the SIMMP–Sepsis framework to advance the field.

## 14. Conclusions

Sepsis is a dynamic immunometabolic disorder that transcends the classical biphasic paradigm of hyperinflammation followed by immunosuppression. Instead, it involves overlapping and persistent immune dysfunctions—including lymphocyte apoptosis, Treg expansion, dendritic and neutrophil impairment, and checkpoint overexpression (e.g., PD-1, CTLA-4). This review supports the concept of immunodynamic disruption, which more accurately reflects the heterogeneous and evolving immune landscape of sepsis. Incorporating this perspective into clinical care requires dynamic immune profiling to guide personalized interventions.

Key biomarkers such as HLA-DR, PD-1/PD-L1, IL-6, IL-10, circulating mtDNA, and epigenetic signatures provide a foundation for stratifying patients according to immune phenotype and therapeutic need. We advocate for routine immune monitoring every 48–72 h in critical care, enabling timely and tailored immunomodulation. This approach may optimize immune balance, reduce harmful over- or under-stimulation, and improve survival and post-ICU outcomes. Importantly, immune-based therapies—such as IL-7, GM-CSF, or checkpoint inhibitors—should be deployed in a phase-specific manner rather than as one-size-fits-all strategies.

At the core of this model lies the Persistent Immune Remnant, a state defined by long-lasting epigenetic, mitochondrial, and metabolic dysfunctions that contribute to post-sepsis morbidity despite apparent clinical recovery. Recognizing sepsis as a persistent immunometabolic syndrome reframes post-acute care from passive observation to proactive immune restoration. The proposed SIMMP–Sepsis model offers a testable and translational scaffold that bridges molecular mechanisms with clinical decision-making, promoting the development of adaptive, biomarker-guided therapies. Ultimately, survival should be seen not as the end of treatment, but as the beginning of long-term immune recovery.

## Figures and Tables

**Figure 1 biomedicines-13-02139-f001:**
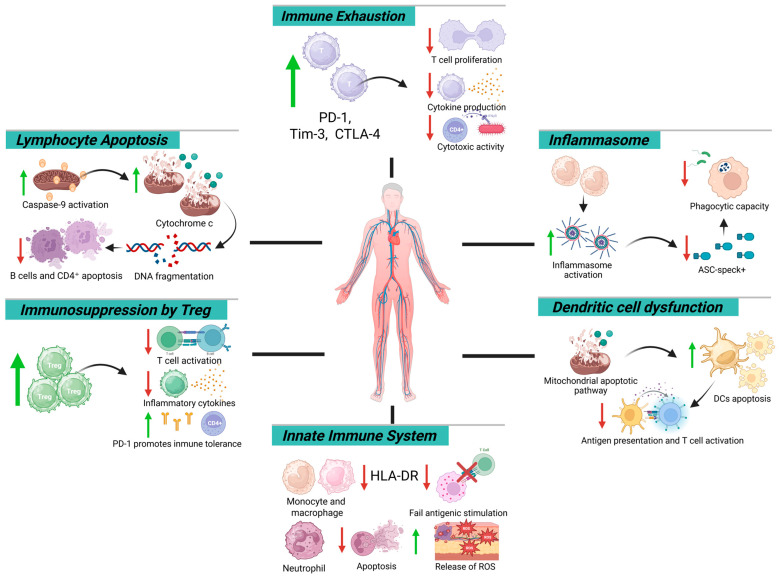
Mechanistic map of immune dysfunction in sepsis: cellular and molecular targets. Green arrows indicate an increase, whereas red arrows indicate a decrease.

**Figure 2 biomedicines-13-02139-f002:**
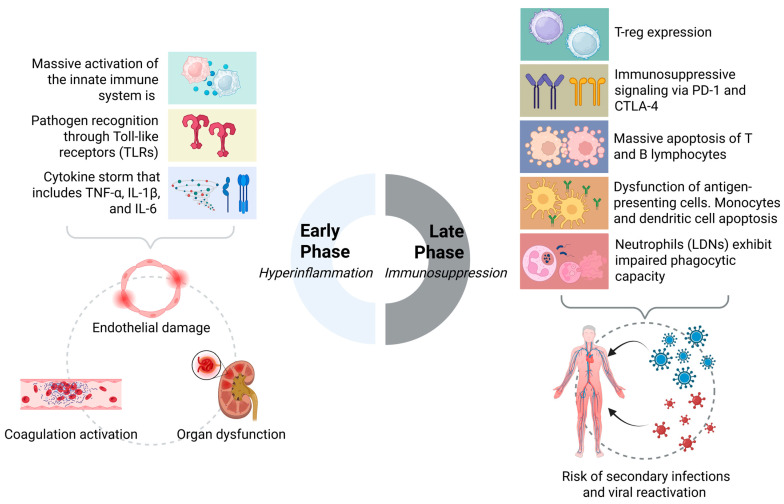
Biphasic immunodynamic response in sepsis: from hyperinflammation to immunosuppression.

**Table 1 biomedicines-13-02139-t001:** Immunological biomarkers in sepsis: clinical utility and limitations.

Biomarker	Source or Cell	Clinical Significance	Limitations/Current Validation	References
HLA-DR	Monocytes	Reduced expression linked to immunosuppression and higher risk of secondary infections.	Variability in measurement techniques and lack of standardized clinical thresholds.	[23,24]
PD-1/PD-L1	T lymphocytes (PD-1) and antigen-presenting cells (PD-L1)	Overexpression correlates with T cell exhaustion and sepsis severity.	Requires flow cytometry; expression varies by disease phase.	[6,25]
IL-10	Immune cells (lymphocytes, macrophages)	Elevated levels indicate immunosuppression and worse prognosis.	Non-specific (elevated in other inflammatory conditions).	[26,27]
mtDNA	Mitochondria released during cellular damage	Activates STING pathway in dendritic cells, associated with immunoparalysis.	Technically challenging to quantify; present in other tissue-damage pathologies.	[28,29]
ASC-specks	Monocytes and neutrophils	Indicator of NLRP3 inflammasome activation; low levels predict higher mortality.	Complex detection (flow cytometry); limited clinical validation.	[30]
LDN	Circulating neutrophils	Associated with immune dysfunction, high PD-L1 expression, and infection risk.	Isolation challenges; heterogeneous studies on prognostic value.	[20,31]
Caspase-3/Caspase-9	Lymphocytes and dendritic cells	Activation indicates apoptosis, contributing to lymphopenia and immunosuppression.	Measuring active caspases is technically challenging in clinical settings.	[23,32]
IL-7R	T lymphocytes	Reduced expression linked to T cell dysfunction; potential therapeutic target.	Experimental-phase studies; variability based on patient immune status.	[6,33]

HLA-DR: Human Leukocyte Antigen–DR isotype, PD-1: programmed cell death protein 1, PD-L1: Programmed Death-Ligand 1, IL-10: Interleukin-10, mtDNA: mitochondrial DNA, STING: Stimulator of Interferon Genes, ASC-specks: Apoptosis-Associated Speck-Like Protein Containing a CARD, NLRP3: NOD-, LRR-, and Pyrin Domain-Containing Protein 3, LDN: low-density neutrophils, Caspase-3: cysteine-aspartic protease 3, Caspase-9: cysteine-aspartic protease 9, IL-7R: Interleukin-7 Receptor.

**Table 2 biomedicines-13-02139-t002:** Key immune cells and their principal dysfunctions in sepsis-induced immunosuppression, separated into adaptive and innate mechanisms and linked to their clinical relevance.

Immune Cell Type	Key Dysregulated Pathways	Functional Outcome (Clinical Relevance)	Reference
**Adaptive Immunity**			
CD4^+^ T lymphocytes	Apoptosis; checkpoint overexpression (PD-1, CTLA-4)	Reduced proliferation, impaired helper function → susceptibility to secondary infections	[6]
CD8^+^ T lymphocytes	Exhaustion, loss of cytotoxicity, PD-1/LAG-3 upregulation	Impaired tumor surveillance, viral reactivation	[50]
B cells	Apoptosis, decreased antibody production	Low antibody levels → poor humoral response, increased reinfection risk	[13]
Regulatory T cells	β1-adrenergic expansion, suppressive dominance	Inhibition of effector T cells → immune tolerance, persistent infection	[51]
**Innate immunity**			
Neutrophils	Delayed apoptosis, PD-L1^+^ low-density neutrophils, impaired chemotaxis	Reduced pathogen clearance, tissue damage	[52]
Monocytes	Decreased HLA-DR, metabolic reprogramming	Loss of antigen presentation → immunoparalysis	[18]
Dendritic cells	Mitochondrial DNA–induced STING activation, apoptosis	Impaired T cell priming, poor adaptive activation	[53]

CD4^+^ T lymphocytes: Cluster of Differentiation 4 positive T cells, PD-1: programmed cell death protein 1, CTLA-4: cytotoxic T lymphocyte–associated protein 4, CD8^+^ T lymphocytes: Cluster of Differentiation 8 positive T cells, LAG-3: Lymphocyte-Activation Gene 3, B cells: B lymphocytes, regulatory T cells (Tregs): CD4^+^CD25^highFoxP3^+^ T cells, β1: Beta-1 adrenergic receptor, Neutrophils: Innate phagocytic granulocytes, PD-L1: Programmed Death-Ligand 1, HLA-DR: Human Leukocyte Antigen–DR isotype, Monocytes: circulating innate immune phagocytes, STING: Stimulator of Interferon Genes.

**Table 3 biomedicines-13-02139-t003:** Innate and adaptive immune dysfunctions in sepsis.

Innate Immune Dysfunction	Reference	Adaptive Immune Dysfunction	Reference
Decreased HLA-DR expression in monocytes	[60]	Massive apoptosis of CD4^+^ and B lymphocytes	[13]
Expansion of low-density neutrophils (LDNs)	[17]	T cell exhaustion (PD-1, LAG-3, CTLA-4 overexpression)	[54]
Dysregulated NLRP3 inflammasome activation	[30]	Reduced IL-7R expression and impaired immune memory	[50]
LPS tolerance and impaired PAMP response	[64]	Th17/Treg axis imbalance	[59]
Elevated circulating mtDNA and STING pathway activation	[53]	Impaired antibody production and humoral response	[13]
Altered neutrophil migration and chemotaxis	[31]	Loss of clonal diversity in T and B cell repertoires	[47]

HLA-DR: Human Leukocyte Antigen–DR isotype, CD4^+^: Cluster of Differentiation 4 positive T cell, B lymphocytes: B cells, LDN: low-density neutrophils, PD-1: programmed cell death protein 1, LAG-3: Lymphocyte-Activation Gene 3, CTLA-4: cytotoxic T lymphocyte–associated protein 4, NLRP3: NOD-, LRR-, and Pyrin Domain-containing Protein 3 (inflammasome sensor), IL-7R: Interleukin-7 Receptor, LPS: lipopolysaccharide, PAMP: Pathogen-Associated Molecular Pattern, Th17: T helper 17 cell, Treg: regulatory T cell, mtDNA: mitochondrial DNA, STING: Stimulator of Interferon Genes.

**Table 4 biomedicines-13-02139-t004:** Immunocellular alterations in the Post-Sepsis Cellular Remnant (PSCR). This table outlines the principal immune and endothelial cell types affected in the PSCR, highlighting their functional impairments, clinical implications, and potential biomarkers for monitoring long-term immune dysfunction after sepsis.

Affected Cell Type	Functional Alterations	Clinical Implications	Biomarkers	Reference
Monocytes	Mitochondrial dysfunction, reduced IL-6, TNF-α, and IFN-γ production.	Leads to immunoparalysis and increased reinfection risk.	↓HLA-DR, ↓TLR5	[60]
CD4^+^ T Lymphocytes	Functional exhaustion.	Favors susceptibility to opportunistic infections.	↑PD-1, ↑BTLA, ↓IFN-γ	[54]
Endothelial Cells	Persistent activation, ↑ ROS, impaired nitric oxide signaling.	Promotes immunothrombosis and microvascular damage.	↑ sVCAM-1, ↑ angiopoietin-2	[105]
Macrophages	Sustained epigenetic reprogramming.	Leads to chronic inflammatory hyper-responsiveness.	↑ H3K4me3, ↑ H3K27ac	[80]
Neutrophils	Persistent NETosis, release of cfDNA, and citrullinated histones.	Drives endothelial injury and thrombotic risk.	cfDNA, citrullinated H3	[70]
CD8^+^ TEMRA T Cells	Senescent phenotype, impaired cytotoxic function.	Reduces immune surveillance.	↓Perforin, ↑CD57^+^	[50]

IL-6: Interleukin-6, TNF-α: Tumor Necrosis Factor alpha, IFN-γ: Interferon gamma, HLA-DR: Human Leukocyte Antigen–DR isotype, TLR5: Toll-Like Receptor 5, PD-1: programmed cell death protein 1, BTLA: B and T Lymphocyte Attenuator, ROS: Reactive Oxygen Species, sVCAM-1: Soluble Vascular Cell Adhesion Molecule 1, Angiopoietin-2: pro-angiogenic growth factor, H3K4me3: trimethylation of histone H3 at lysine 4, H3K27ac: acetylation of histone H3 at lysine 27, NETosis: Neutrophil Extracellular Trap formation, cfDNA: Cell-free DNA, TEMRA: Terminally Differentiated Effector Memory CD8^+^ T cell re-expressing CD45RA, Perforin: cytolytic protein involved in target cell lysis, CD57^+^: differentiation marker of senescent or terminally differentiated T cells. Upward arrows (↑) indicate an increase, whereas downward arrows (↓) indicate a decrease.

**Table 5 biomedicines-13-02139-t005:** Summary of the main experimental models of immunoparalysis in sepsis. This table describes an overview of the principal models used to investigate immune collapse in sepsis. Each model replicates specific aspects of immunoparalysis, with implications for translational research and immunotherapeutic development.

Model	Description	Strengths	Limitations	References
One-hit model	Single septic insults using LPS, live bacteria, or CLP to induce sepsis.	Reproduces early immune alterations such as T cell apoptosis and MDSC/Treg expansion.	Fails to replicate transition to prolonged immunosuppression.	[60,67]
Two-hit model	Initial sepsis followed by a secondary infection (e.g., *P. aeruginosa*, *C. albicans*).	Mimics clinical progression; induces PD-1/PD-L1 upregulation and impaired lymphocyte proliferation.	High variability in secondary insult and timing; challenges for standardization.	[66,114]
LPS tolerance	Repeated sublethal LPS doses induce hyporesponsive leukocyte phenotype.	Useful to study leukocyte reprogramming and cytokine suppression.	Does not reflect polymicrobial or systemic infection features of human sepsis.	[64]

LPS: lipopolysaccharide, CLP: cecal ligation and puncture, MDSC: myeloid-derived suppressor cell, Treg: regulatory T cell, PD-1: programmed cell death protein 1, PD-L1: Programmed Death-Ligand 1, *P. aeruginosa*: *Pseudomonas aeruginosa*, *C. albicans*: *Candida albicans*.

**Table 6 biomedicines-13-02139-t006:** Hierarchical cascade of the SIMMP–Sepsis model. This table shows progressive pathophysiological stages from acute immune activation to sustained immunometabolic dysfunction and clinical deterioration in sepsis survivors.

Level	Description	Key Pathophysiological Processes	Clinical Consequences	Reference
1. Acute Activation	Initial immune response to PAMPs and DAMPs	IL-1β, TNF, NETosis, complement activation, early mitochondrial dysfunction, redox imbalance, endothelial injury	Acute multiorgan dysfunction, need for life support, ICU admission	[1]
2. Cellular Reprogramming	Persistent molecular alterations beyond clinical recovery	Trained immunity, maladaptive tolerance, epigenetic marks (H3K4me3, histone lactylation), Warburg-like metabolism, lactate and succinate accumulation	Sustained immunometabolic activation, relapsed risk, subclinical inflammation	[18]
3. Prolonged Organ Dysfunction	Multisystemic consolidation of injury	Mitochondrial fragmentation, mtDNA release, incomplete mitophagy, elevated ICAM-1/VCAM-1, chronic neuroinflammation, gut dysbiosis	Persistent endotheliopathy, neurodysfunction, inflammatory liver injury, increased intestinal permeability	[99]
4. Post-Sepsis Clinical Phenotypes	Dynamic clinical manifestations of persistent damage	Fatigue, myalgia, dysautonomia, cognitive decline, recurrent infections, multiorgan fibrosis, delayed mortality	Functional decline, frequent rehospitalizations, chronic disability, reduced quality of life	[128]
5. Integrated Clinical Entity	Diagnostic proposal: SIMMP–Sepsis	Interaction across 5 domains: innate immunity, cellular metabolism, endothelium, neuroimmune axis, intestinal microbiota	Phenotypic stratification, extended monitoring, targeted immunometabolic intervention	[135]

PAMPs: Pathogen-Associated Molecular Patterns, DAMPs: Damage-Associated Molecular Patterns, IL-1β: Interleukin-1 beta, TNF: Tumor Necrosis Factor, NETosis: Neutrophil Extracellular Trap formation, ICU: Intensive Care Unit, H3K4me3: trimethylation of histone H3 at lysine 4, Histone lactylation: post-translational modification of histones via lactate addition, mtDNA: mitochondrial DNA, ICAM-1: Intercellular Adhesion Molecule 1, VCAM-1: Vascular Cell Adhesion Molecule 1, SIMMP–Sepsis: Sepsis-Associated Persistent Multiorgan Immunometabolic Syndrome.

**Table 7 biomedicines-13-02139-t007:** Summary of the biphasic immune response in sepsis. This table shows that the immune response in sepsis evolves from an initial hyperinflammatory surge to prolonged immunosuppressive dysfunction, shaping the timing and effectiveness of targeted immunotherapies.

Component	Description	Reference
Immune Activation Timeline	Immune response in sepsis follows a biphasic pattern: an early hyperinflammatory phase (≤72 h) with cytokine storm and tissue damage, followed by a prolonged immunosuppressive phase (after day 7) marked by immune cell exhaustion and impaired pathogen clearance.	[60]
Predominant Immune Cells	Early phase: neutrophils, macrophages, and natural killer cells. Late phase: exhausted CD4^+^/CD8^+^ T cells, monocytes with low HLA-DR expression, dysfunctional dendritic cells.	[14]
Key Cytokines and Mediators	Proinflammatory: IL-1β, TNF-α, IL-6, ROS, NETs. Anti-inflammatory: IL-10, TGF-β; Immunosuppressive: PD-1, CTLA-4, BTLA expression, persistent lymphopenia.	[60]
Molecular Dysregulation	NETosis, mitochondrial dysfunction, glycolytic shift, upregulation of immune checkpoints, and epigenetic reprogramming of immune cells.	[18]
Clinical Manifestations	Early: fever, hypotension, ARDS, and multiorgan failure. Late: secondary infections, viral reactivation, impaired wound healing, prolonged ICU stay, and late mortality.	[59]
Diagnostic Biomarkers	Elevated IL-6 and sTNFR1, circulating mitochondrial DNA, reduced HLA-DR on monocytes, persistent PD-1 on T cells, and S100A8/A9.	[77]
Therapeutic Windows	The transition phase (days 3–7) is optimal for immune profiling and tailored interventions. Immunotherapies should be avoided during the hyperinflammatory phase and initiated during adaptive immune suppression.	[44]
Potential Interventions	Early: source control, antibiotics, fluids, and vasopressors. Late: recombinant IL-7, anti–PD-1 antibodies, statins, antioxidants, metabolic modulators to restore immune function and prevent complications.	[33]

CD4^+^: Cluster of Differentiation 4 positive T cell, CD8^+^: Cluster of Differentiation 8 positive T cell, HLA-DR: Human Leukocyte Antigen–DR isotype, IL-1β: Interleukin-1 beta, TNF-α: Tumor Necrosis Factor alpha, IL-6: Interleukin-6, ROS: Reactive Oxygen Species, NETs: Neutrophil Extracellular Traps, IL-10: Interleukin-10, TGF-β: transforming growth factor beta, PD-1: programmed cell death protein 1, CTLA-4: cytotoxic T lymphocyte–associated protein 4, BTLA: B and T Lymphocyte Attenuator, ARDS: Acute Respiratory Distress Syndrome, ICU: Intensive Care Unit, sTNFR1: Soluble Tumor Necrosis Factor Receptor 1, mtDNA: mitochondrial DNA, S100A8/A9: Calprotectin (heterodimer of S100A8 and S100A9 proteins), IL-7: Interleukin-7.

**Table 8 biomedicines-13-02139-t008:** Conceptual evolution of immunological models in sepsis and the clinical value of dynamic immunoprofiling.

Immunological Paradigms in Sepsis	Core Description	Limitations	Added Value of the Immunodynamic Disruption Model	Reference
Classic Paradigm (Hyperinflammation → Immunosuppression)	Sequential model with an early proinflammatory phase followed by immunosuppression.	Oversimplifies the timeline; does not account for overlapping responses or cell-specific contradictions.	Recognizes immune heterogeneity, temporal overlap, and dual-function immune cells.	[60]
Previous Models (Immunoparalysis, Trained Immunity)	Describe immune tolerance or epigenetically driven immune enhancement.	Do not integrate coexisting immune states or account for time-dependent transitions.	Provide partial frameworks that are unified in the immunodynamic model.	[98]
Proposed Model (Immunodynamic Disruption + Persistent Immune Remnant) [44]	Highlights dynamic coexistence of immune activation and suppression, driven by epigenetic, functional, and metabolic rewiring.	Remains under clinical validation.	Introduces the concept of a Persistent Immune Remnant to explain lasting immunometabolic dysfunction and supports stratified, biomarker-guided decision-making.	[44]

## Data Availability

Not applicable.

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
