# Peer review of "Immunodynamic Disruption in Sepsis: Mechanisms and Strategies for Personalized Immunomodulation"

_biomedicines, 2025, doi:10.3390/biomedicines13092139_

Round 1

Reviewer 1 Report

Comments and Suggestions for Authors

I reviewed the manuscript entitled Immunodynamic Disruption in Sepsis: Mechanisms,

Trajectories, and Pathways to Personalized Immunomodulation—A Comprehensive Review

  1. Immunology of sepsis is an important issue, and the topic is well selected.
  2. Please explain about the S100A8/A9. Replace with the common name.
  3. How can clinicians discriminate the early phase of sepsis (hyperinflammation) and late phase (immunosuppression)?
  4. This review is long. Please make it shorter if possible.

Author Response

Dear Reviewers,
We sincerely thank you for your thorough evaluation and constructive feedback aimed at improving our manuscript. Below, we provide a detailed point-by-point response addressing each of your comments. In addition, we have submitted a revised version of the manuscript in which all changes are highlighted in a different font color to facilitate their identification.

Point by point letter

Re: “Immunodynamic Disruption in Sepsis: Mechanisms, Trajectories, and Pathways to Personalized Immunomodulation — A Comprehensive Review”

Reviewer 1

I reviewed the manuscript entitled Immunodynamic Disruption in Sepsis: Mechanisms,

Trajectories, and Pathways to Personalized Immunomodulation—A Comprehensive Review

  1. Immunology of sepsis is an important issue, and the topic is well selected.

Thanks for your comment.

  1. Please explain about the S100A8/A9. Replace with the common name.

Thank you for this suggestion. We have replaced “S100A8/A9” with its common name “Calprotectin” throughout the manuscript and clarified its role in endothelial disruption.

  1. How can clinicians discriminate the early phase of sepsis (hyperinflammation) and late phase (immunosuppression)?

Thank you for this important observation. In clinical practice, these phases can be differentiated by combining bedside manifestations with immunological biomarkers. We have incorporated this clarification into the manuscript to strengthen its translational value.

  1. This review is long. Please make it shorter if possible.

Thank you for your recommendation. We carefully revised the manuscript, removing unnecessary text and redundancies. We also reorganized and streamlined several sections to improve clarity and conciseness.

Reviewer 2 Report

Comments and Suggestions for Authors

Title

The title is long. Redundant and unclear phrases like “Comprehensive Review” and “Pathways to Personalized Immunomodulation”  should be deleted since the manuscript is already marked as a review by the journal. I suggest the authors consider changing the title to “Immunodynamic disruption in sepsis: Mechanisms and strategies for personalized immunomodulation”

Abstract

A non-expert may find it hard to follow because of overuse of dense sentences and multiple concepts. Phrases like “Persistent Immune Remnant” should be defined briefly in plain terms. Many abbreviations such as PD-1, CTLA-4, SIMPP etc. were not defined first time mentioned, making it confusing. Only important biomarkers should be mentioned

27-28: State the findings of the study rather than the experiments done

Introduction

There are long sentences which should be broken in lines  46–50 and 52–55

67-68: How does “Immunodynamic disruption” contrast with other established terms?

Immunodynamic Disruption in Sepsis

Authors repeated “Immunodynamic disruption” many times but made no advances in explaining the concept thereby making it redundant

71: Back  this with evidence “more accurately described as” 

All Tables should have a separate column for Reference and all the abbreviations should be defined as a legend under the table

In table 2, some cell function outcomes were mentioned in earlier sections making it redundant

Can  “hypogammaglobulinemia” in table 2 be replaced with “low antibody levels” to enhance comprehension?

Adaptive and innate mechanisms should be separated and each mechanism linked with its clinical relevance

216: It is confusing whether “Post-Sepsis Cellular Remnant” or “Persistent Immune Remnant” mentioned earlier in the abstract, is being discussed. Clarify this inconsistency

Table 3 title is overly long, summarize it by removing the technical terms

132-178: How can each of the alteration guide monitoring or treatment?

Translational Value of Experimental Models

The model limitations were sufficiently discussed, but how to bridge the gap to human studies was overlooked

In a paragraph, clarify recommendations for standardization/how to improve translational relevance

Please add a table comparing the utility of each model

Clinical Implications

539-562: The discussion on IL-36 shows inconclusive evidence, but authors presented it with strong associations

711-728: “Precision Immunotherapy” section is similar to the discussion on checkpoint blockade

Discussion came at the end of the review whereas this is not a systematic review. I think Conclusion should be the appropriate ending since it is a narrative review

State the limitations of the review?

Typographical errors dotted the entire manuscript

Comments on the Quality of English Language

Minor editing is required

Author Response

Dear Reviewers,
We sincerely thank you for your thorough evaluation and constructive feedback aimed at improving our manuscript. Below, we provide a detailed point-by-point response addressing each of your comments. In addition, we have submitted a revised version of the manuscript in which all changes are highlighted in a different font color to facilitate their identification.

Point by point letter

Re: “Immunodynamic Disruption in Sepsis: Mechanisms, Trajectories, and Pathways to Personalized Immunomodulation — A Comprehensive Review”

Reviewer 2

Title

The title is long. Redundant and unclear phrases like “Comprehensive Review” and “Pathways to Personalized Immunomodulation”  should be deleted since the manuscript is already marked as a review by the journal. I suggest the authors consider changing the title to “Immunodynamic disruption in sepsis: Mechanisms and strategies for personalized immunomodulation”

-Thank you for this valuable suggestion. The title has been revised accordingly.

Abstract

A non-expert may find it hard to follow because of overuse of dense sentences and multiple concepts. Phrases like “Persistent Immune Remnant” should be defined briefly in plain terms. Many abbreviations such as PD-1, CTLA-4, SIMPP etc. were not defined first time mentioned, making it confusing. Only important biomarkers should be mentioned

27-28: State the findings of the study rather than the experiments done

-Thank you for your comment. We revised the abstract to focus on the findings of the review, defined key terms (e.g., PD-1, CTLA-4, SIMMP) upon first mention, and removed unnecessary abbreviations. Concepts such as the “Post-Sepsis Cellular Remnant” are now briefly explained in plain terms.

Introduction

There are long sentences which should be broken in lines  46–50 and 52–55

-Thank you for this helpful observation. We revised the introduction to break down long sentences and improve readability.

67-68: How does “Immunodynamic disruption” contrast with other established terms?

-We appreciate this important comment. We clarified that unlike immunoparalysis (a unidirectional progression toward suppression) or trained immunity (focused on innate memory), immunodynamic disruption describes the coexistence of hyperactivation, exhaustion, and tolerance across innate and adaptive compartments. This broader framework better explains persistent immune dysfunction and long-term sequelae of sepsis.

Immunodynamic Disruption in Sepsis

Authors repeated “Immunodynamic disruption” many times but made no advances in explaining the concept thereby making it redundant

71: Back  this with evidence “more accurately described as” 

-Thank you for this observation. We now support this statement with additional references, which demonstrate that sepsis is more accurately described as an evolving and heterogeneous immunodynamic process rather than as a simple state of immunoparalysis.

All Tables should have a separate column for Reference and all the abbreviations should be defined as a legend under the table

-Thank you. We have corrected all tables accordingly.

In table 2, some cell function outcomes were mentioned in earlier sections making it redundant

-Thank you. We revised Table 2 to remove redundancies.

Can  “hypogammaglobulinemia” in table 2 be replaced with “low antibody levels” to enhance comprehension?

-Thank you. We replaced the term as suggested.

Adaptive and innate mechanisms should be separated and each mechanism linked with its clinical relevance

-Thank you. We reorganized the section to clearly separate adaptive and innate mechanisms and linked each to its clinical implications, improving clarity and translational value.

216: It is confusing whether “Post-Sepsis Cellular Remnant” or “Persistent Immune Remnant” mentioned earlier in the abstract, is being discussed. Clarify this inconsistency

-Thank you for noting this. We ensured consistency by using the term “Post-Sepsis Cellular Remnant (PSCR)” throughout the manuscript, including the abstract.

Table 3 title is overly long, summarize it by removing the technical terms

-Thank you. We shortened the title for clarity.

132-178: How can each of the alteration guide monitoring or treatment?

-Thank you. We specified in the manuscript how each alteration can inform patient monitoring or therapeutic strategies.

Translational Value of Experimental Models

The model limitations were sufficiently discussed, but how to bridge the gap to human studies was overlooked

In a paragraph, clarify recommendations for standardization/how to improve translational relevance

Please add a table comparing the utility of each model

-Thank you for pointing this out. We expanded this section by adding a dedicated paragraph that outlines strategies to improve translational relevance, including standardization through MQTiPSS guidelines, incorporation of clinically relevant variables (age, sex, comorbidities), and alignment of experimental endpoints with human outcomes. In addition, we created a new comparative table (Table X) summarizing the strengths, limitations, and translational value of each model.

Clinical Implications

539-562: The discussion on IL-36 shows inconclusive evidence, but authors presented it with strong associations

-Thank you. We revised this section to reflect the inconclusive nature of the evidence and avoid overstating associations.

711-728: “Precision Immunotherapy” section is similar to the discussion on checkpoint blockade

-We agree with this observation. We revised the text to reduce redundancy and clarify the distinction between checkpoint blockade and broader precision immunotherapy.

Discussion came at the end of the review whereas this is not a systematic review. I think Conclusion should be the appropriate ending since it is a narrative review

- Thank you for this comment. We reorganized the manuscript to improve clarity and flow, and we now conclude with a dedicated Conclusions section following the discussion and limitations.

State the limitations of the review?

-Thank you. We added a dedicated subsection on the limitations of this review at the end of the Discussion.

Typographical errors dotted the entire manuscript

-Thank you. We carefully proofread and corrected all typographical errors.

Reviewer 3 Report

Comments and Suggestions for Authors

The authors provide an interesting review focusing on personalized interventions in sepsis. This perspective is particularly timely with the recent expansion of knowledge on personalized medicine, and the continued lack of sepsis-specific treatments, contributing to high mortality. 

My comments are as follows. 

Introduction:

  • The introduction is concise and provides a sound overview of sepsis from an immune perspective. It would help to add some epidemiological background to the introduction to give the reader an idea of the scope of the problem. 

Experimental models: 

  • Models of sepsis and immunoparalysis could be combined into one section. The first section initially appears somewhat weak, but it is later expanded in the immunoparalysis models section. 
  • Cecal slurry is currently in use by many groups due to its reproducibility, and should be mentioned. 

Table 7: 

  • This is a very informative table, but might work better earlier in the manuscript as it largely provides descriptions of different elements of the immune response to sepsis. 

General comments:

  • The SIMMP framework is interesting, and merits a level 1 heading, rather than a subheading. This to me is the novel part of the manuscript. 
  • The overall structure of the manuscript does not seem to have a logical flow. I encourage the authors to reconsider the headings and subheadings, ensuring there is clear flow throughout the manuscript. Related concepts can be groups under one heading, with some accompanying text laying out the overall concept and effectively justifying including the subheadings under this heading. The different portions can then go under subheadings. 

Author Response

Dear Reviewers,
We sincerely thank you for your thorough evaluation and constructive feedback aimed at improving our manuscript. Below, we provide a detailed point-by-point response addressing each of your comments. In addition, we have submitted a revised version of the manuscript in which all changes are highlighted in a different font color to facilitate their identification.

Point by point letter

Re: “Immunodynamic Disruption in Sepsis: Mechanisms, Trajectories, and Pathways to Personalized Immunomodulation — A Comprehensive Review”

Reviewer 3

The authors provide an interesting review focusing on personalized interventions in sepsis. This perspective is particularly timely with the recent expansion of knowledge on personalized medicine, and the continued lack of sepsis-specific treatments, contributing to high mortality. 

My comments are as follows. 

Introduction:

  • The introduction is concise and provides a sound overview of sepsis from an immune perspective. It would help to add some epidemiological background to the introduction to give the reader an idea of the scope of the problem. 

Thank you. We added epidemiological background to better contextualize the global burden of sepsis.

Experimental models: 

  • Models of sepsis and immunoparalysis could be combined into one section. The first section initially appears somewhat weak, but it is later expanded in the immunoparalysis models section. 

Thank you. We reorganized this section into a unified framework combining sepsis and immunoparalysis models.

  • Cecal slurry is currently in use by many groups due to its reproducibility, and should be mentioned. 

Thank you. We added discussion of the cecal slurry model, which is increasingly used for its reproducibility.

Table 7: 

  • This is a very informative table, but might work better earlier in the manuscript as it largely provides descriptions of different elements of the immune response to sepsis. 

Thanks for your comment. We considered this suggestion but decided to retain Table 7 in its current position, as it directly supports the surrounding sections. However, we are open to relocating it if the editor believes it would improve readability.

General comments:

  • The SIMMP framework is interesting, and merits a level 1 heading, rather than a subheading. This to me is the novel part of the manuscript. 

Thanks for your suggestion. We elevated SIMMP–Sepsis to a level-1 heading and expanded its discussion to highlight its novelty.

  • The overall structure of the manuscript does not seem to have a logical flow. I encourage the authors to reconsider the headings and subheadings, ensuring there is clear flow throughout the manuscript. Related concepts can be groups under one heading, with some accompanying text laying out the overall concept and effectively justifying including the subheadings under this heading. The different portions can then go under subheadings. 

Thank you for this constructive suggestion. We carefully revised the headings and subheadings to improve logical flow. Related concepts are now grouped under broader section titles with introductory text explaining their integration (e.g., adaptive and innate immune dysfunctions are clearly separated but presented under a common framework of sepsis-induced immunosuppression). In addition, we elevated SIMMP–Sepsis to a level-1 heading to emphasize its novelty and centrality. These adjustments provide a clearer narrative progression from mechanisms, to models, to clinical implications.